# Comparison between Automatic and Semiautomatic System for the 3D Echocardiographic Multiparametric Evaluation of RV Function and Dimension

**DOI:** 10.3390/jcm11154528

**Published:** 2022-08-03

**Authors:** Marco Penso, Remo Antonio Ranalletta, Mauro Pepi, Anna Garlaschè, Sarah Ghulam Ali, Laura Fusini, Valentina Mantegazza, Manuela Muratori, Riccardo Maragna, Gloria Tamborini

**Affiliations:** 1Imaging Department, Centro Cardiologico Monzino IRCCS, 20138 Milan, Italy; remoantonio_ranalletta@hotmail.it (R.A.R.); mauro.pepi@cardiologicomonzino.it (M.P.); anna.garlasche@cardiologicomonzino.it (A.G.); sarah.ghulamali@cardiologicomonzino.it (S.G.A.); laura.fusini@cardiologicomonzino.it (L.F.); valentina.mantegazza@cardiologicomonzino.it (V.M.); manuela.muratori@cardiologicomonzino.it (M.M.); riccardo.maragna@unimi.it (R.M.); gloria.tamborini@cardiologicomonzino.it (G.T.); 2Department of Electronics, Information and Biomedical Engineering, Politecnico di Milano, 20133 Milan, Italy

**Keywords:** right ventricle, three-dimensional echocardiography, artificial intelligence, right ventricle volumes, right ventricle ejection fraction, automatic analysis

## Abstract

Background: The right ventricle (RV) plays a pivotal role in cardiovascular diseases and 3-dimensional echocardiography (3DE) has gained acceptance for the evaluation of RV volumes and function. Recently, a new artificial intelligence (AI)–based automated 3DE software for RV evaluation has been proposed and validated against cardiac magnetic resonance. The aims of this study were three-fold: (i) feasibility of the AI-based 3DE RV quantification, (ii) comparison with the semi-automatic 3DE method and (iii) assessment of 2-dimensional echocardiography (2DE) and strain measurements obtained automatically. Methods: A total of 203 subject (122 normal and 81 patients) underwent a 2DE and both the semi-automatic and automatic 3DE methods for Doppler standard, RV volumes and ejection fraction (RVEF) measurements. Results: The automatic 3DE method was highly feasible, faster than 2DE and semi-automatic 3DE and data obtained were comparable with traditional measurements. Both in normal subjects and patients, the RVEF was similar to the two 3DE methods and 2DE and strain measurements obtained by the automated system correlated very well with the standard 2DE and strain ones. Conclusions: results showed that rapid analysis and excellent reproducibility of AI-based 3DE RV analysis supported the routine adoption of this automated method in the daily clinical workflow.

## 1. Introduction

The echocardiographic study of the function and size of the right ventricle (RV) has always been of great interest for its diagnostic and prognostic role in many cardiac pathologies but has been hindered by the extremely complex structure of the RV, both from a morphological and functional point of view. Ultrasound attempts to quantify volumes and ejection fractions using appropriate 2-dimensional (2D) geometric models for RV evaluation have failed in studies comparing cardiac magnetic resonance (CMR) imaging. The 3-dimensional (3D) echocardiographic (3DE) method overcoming these limits has allowed a direct quantification of the RV volumes and ejection fraction (RVEF) in the absence of geometric assumptions [1,2]. A comprehensive assessment of RV performance should also include a measure of RV size, right atrial (RA) size, RV systolic function and at least one of the following: fractional area change (FAC), tricuspid annular plane systolic excursion (TAPSE), peak systolic velocity (PSV) and systolic pulmonary artery pressure (SPAP), with estimate of RA pressure on the basis of inferior vena cava size and collapse [3,4]. More recently, RV global and free wall longitudinal strain (FWLS) were included in the evaluation of RV function, thanks to their clinical value over conventional measurements to detect subtle RV systolic abnormalities [5].

In the last few years, 3DE quantification of RV has been proposed and validated in many clinical settings [6,7,8,9]. However, the necessity of complicated on-board or offline dedicated software has reduced the diffusion of 3DE RV evaluation in daily practice. To overcome this limitation, new simplified “on-board” 3D software has been developed to obtain RV volumes and RVEF, together with some of the most used traditional parameters of RV function, automatically derived from 3DE: TAPSE, FAC and FWLS [10]. More recently, a new artificial intelligence (AI)–based 3DE software for automatic RV size and functional quantification has been proposed and validated against CMR [11]. This new on-board 3D software not only allowed automatic 3DE measurements in less than 20 s, but also a simultaneous analysis of several 2D echocardiographic (2DE) parameters.

The aims of this study were three-fold: (a) to evaluate the feasibility of an AI-based automated system for 3D assessment of RV volumes and function in a large unselected population of normal and pathological subjects; (b) to compare the automated AI-based system with the standard semi-automatic 3D method; and (c) to assess also, with the new system, a comprehensive 2D echo-Doppler and strain measurements and correlate the results with the conventional 2DE functional parameters.

## 2. Materials and Methods

### 2.1. Study Population and Design

The study population consisted of 215 consecutive patients referred for transthoracic echocardiography in our laboratory. Exclusion criteria were the presence of inadequate echocardiographic apical window.

Of the 215 cases 127 were normal controls with no echocardiographic evidence of heart or valvular disease who underwent transthoracic echocardiography for atypical chest pain, palpitations or innocent cardiac murmurs. On the contrary, 88 patients had clinical or echocardiographic evidence of cardiac disease: valvular heart disease (39 cases), coronary artery disease (14 cases), idiopathic dilated cardiomyopathy (15 cases), congenital or acquired pathologies associated with RV pressure or volume overload (20 cases).

Each patient underwent a complete 2DE and 3DE examination. The study protocol conformed to the ethical guidelines of the 1975 Declaration of Helsinki as reflected in a priori approval by the institution’s human research committee and was approved by the institutional review board and Ethical Committee. Informed consent was obtained from each patient.

### 2.2. Two-Dimensional Echocardiography

The standard comprehensive echocardiographic examinations were performed using an EPIQ 7C echocardiographic system (Philips Healthcare, Andover, MA, USA), equipped with an X5-1 phased-array transducer. According to clinical laboratory practice left ventricular (LV) ejection fraction was calculated using the Simpson’s formula measuring LV end-diastolic and end-systolic volumes from the 4- and 2-chamber views. LV stroke volume was measured as the difference in the volumes. 

As concerns the RV parameters in all cases FAC, TAPSE and FWLS were measured according to international guidelines [1]. SPAP was calculated from the RV-RA gradient with estimate of RA pressure on the basis of inferior vena cava size and collapse [12].

### 2.3. Three-Dimensional Echocardiography

The standard 3DE was performed at the end of the 2DE. In brief, a full-volume 3DE dataset of the RV using a X5-1 matrix transducer (Philips Healthcare) with the transducer positioned to obtain a 2D RV focused view (generally with a more lateral position of the transducer) was acquired.

The datasets were stored on the ultrasound machine and the image datasets were transferred to a separate workstation for later analysis. Specifically, 3DE datasets were exported in DICOM format to another workstation to measure RV volumes and RVEF using a commercially available semi-automated vendor-independent 3DE RV quantification software (RV-Function, TomTec Imaging Systems GmbH, Unterschleissheim, Germany). The methodologies for the RV quantification have been previously reported [2,9]. When RV endocardial contours were suboptimal the operator could intervene and adjust the borders manually.

Finally, in the same view the AI acquisition mode was activated to acquire a one-beat full-volume 3DE dataset focusing on the RV and RA. The system acquires a focused one-beat 3DE full volume and datasets were analyzed on board (immediately after the acquisition or at the end of the examination) in few seconds using a novel automated RV quantification software (3D Auto RV, Philips Medical Systems) that detects RV endocardial surfaces using AI, which consists of knowledge-based identification of initial global shape and RV chamber orientation, followed by 3D speckle tracking analysis throughout one cardiac cycle [11]. The software constructed a 3D endocardial cast of the RV at the end-diastole by the Heart Model segmentation algorithm. Subsequently, the software performed 3D speckle tracking analysis on the RV endocardial border and provided RV end-diastolic volume (RVEDV), RV end-systolic volume (RVESV), RV stroke volume (RVSV) and RVEF providing a tracing of the RV functional curve. When RV endocardial contours were considered suboptimal the operator could intervene manually, fine-tuning the endocardial surface performed interactively to optimize boundary position as necessary.

Moreover, with a simple click on a dedicated icon that appears with the same reconstructed 3D imaging, 2D measurements were also automatically produced including FAC, TAPSE and also including the septum and free wall or only the FWLS.

Figure 1 shows an example of the acquisition and step-by-step analysis obtained with this automated system.

Mean time analysis of both the semi-automatic and automated AI-based system was also annotated by measuring the time from the launch of the acquired data to the appearance of the measurements. Moreover, in cases with the need for editing, the time for analysis was calculated.

### 2.4. Statistical Analysis

Statistical analysis was conducted in SPSS, version 27.0 (SPSS Inc., Chicago, IL, USA). Continuous variables, reported as mean ± standard deviation, were compared with the Student’s unpaired *t*-test. Intra-class correlation coefficient (ICC) and Bland–Altman analysis were used to compare automatic 3DE RV volumes and RVEF with 3DE semi-automatic values. ICC and Bland–Altman analysis were performed also between automatic 2DE-derived FAC, TAPSE, FWLS and the conventional 2DE-derived corresponding functional parameters. To compare the mean time analysis values between the automatic 3DE RV volumes with and without manual correction and the semi-automatic 3DE RV measurements, repeated measures one-way ANOVA test with the Bonferroni correction was utilized. A *p* value < 0.05 was considered statistically significant. Confidence intervals (CI) were set at 95%.

Data for test–retest reliability of the automatic 3DE RV volumes were obtained by the same operator (G.T.) in 20 randomly chosen patients by removing the probe after the first acquisition and repositioning the transducer after 5 min to obtain the second dataset. Further, 3DE RV volumes were automatically obtained without operator intervention. To assess the reproducibility of the semi-automatic 3DE RV measurements in a subset of 20 randomly chosen patients, the first operator (G.T.) reevaluated the same 3D datasets 2 weeks after the first analysis, blinded with respect to the results of the previous evaluation. For each computed parameter, intra-observer variability was then evaluated. The same subset was also evaluated by a second observer (A.R.), blinded to the results obtained by G.T, to assess inter-observer variability. Both variability and test–retest analysis were expressed in terms of ICC and Bland–Altman analysis. Good reproducibility was indicated by an ICC > 0.75 between measurements.

## 3. Results

At least one good 3DE acquisition of the RV was achieved in all 215 subjects; however, despite sufficient acquisition, the quality of 3DE RV reconstruction (both with semi-automatic and automated methods) was insufficient in 12 patients (5 normal cases, 2 dilated cardiomyopathies, 2 RV disease with RV volume overload and 3 coronary artery disease). Thus, the feasibility of RV reconstruction was 94.5% and our final study population consisted of 203 subjects: 122 normal (group 1) and 81 pathological cases (group 2). Atrial fibrillation was present in 10 cases in whom the feasibility was 100%.

The demographic and 2D transthoracic echocardiography characteristics of the study population are reported in Table 1. Pathological patients were older than normal subjects, with similar body surface area and higher values of 2D LV, left atrial volumes and had lower LV ejection fractions. 

Table 2 shows the comparison of RV volumes and RVEF with the automated AI-based 3DE method and the semi-automatic one in the entire population and in the two subgroups. Despite semi-automatic volumes, measurements were slightly but significantly higher and differences were negligible from a clinical point of view. In patients, the end-diastolic RV volumes were slightly overestimated with the automated method but the bias versus the traditional semi-automatic one was only 7.3 mL. Bias for the end-systolic volume was even lower (2.8 mL). Interestingly the RVEF was similar to the two methods in normal subjects and patients. 

As concerns 2D data, Table 3 reports values and correlations between automatic AI and standard 2D parameters of the RV. FWLS, TAPSE and FAC showed statistically minimal differences with the two methods (i.e., automatic versus semi-automatic) and, particularly, in patients, these differences were not clinically relevant.

Figure 2 shows Bland–Altman analysis for the main data obtained with the automatic versus semi-automatic methods.

Analysis time for the two 3DE systems and 2DE measurements clearly showed that the automated method was significantly faster than the corresponding semi-automatic method and standard 2D measurements. The automated method was considered optimal without the need for manual adjustments in 55/203 (27%) patients with analysis time of 15 ± 2 s. In the other 148 cases (73%), with suboptimal RV contour, endocardial contour editing was necessary after the automated postprocessing, prolonging analysis time to 100 ± 16 s. The mean time for analysis with the semi-automatic software was 122 ± 13 s, significantly longer (*p* < 0.001) than the automated method. These data are similar results to those of previous work [11].

### Reproducibility

Intra-observer variability in the 3DE measurements of RVEDV, RVESV and RVEF evaluated with ICC were 1.00 (*p* < 0.001, 95%CI: 0.99–1.00), 0.99 (*p* < 0.001, 95%CI: 0.98–1.00) and 0.92 (*p* < 0.001, 95%CI: 0.90–0.98), respectively, while inter-observer variability of the same measures was 0.98 (*p* < 0.001, 95%CI: 0.96–0.99), 0.91 (*p* < 0.001, 95%CI: 0.79–0.97) and 0.58 (*p* = 0.032, 95%CI: 0.35–0.83), respectively. When evaluating the test–retest reliability of the same clinical parameters with ICC, high values of reproducibility were observed (ICC: RVEDV, 0.99, *p* < 0.001, 95%CI: 0.96–0.99; RVESV, 0.99, *p* < 0.001, 95%CI: 0.97–1.00; RVEF, 0.95, *p* < 0.001, 95%CI: 0.88–0.98). The results of Bland–Altman analysis of the agreement between repeated measurements of RVEDV, RVESV and RVEF, together with the relevant biases and limits of agreement, are depicted in Figure 3.

Figure 4 andFigure 5 show examples of three cases in whom automated AI-based system where applied. Specifically, an atrial septal defect, a volume overload and a pressure overload patient were depicted.

## 4. Discussion

The main findings according to the aims of the study were: (a) an on-board AI-based software for 3D assessment of RV volume and function is very fast, highly feasible and may be utilized in a routine busy echo laboratory; (b) data obtained with this AI-based system are comparable to the standard semi-automated 3D system; and (c) the AI-based system not only allowed 3D analysis but also a comprehensive 2D evaluation of conventional functional parameters in a few seconds.

RV size and function measurements are important for the diagnosis and prognostic evaluation of multiple cardiac diseases. Several studies have established the accuracy of 3DE measurements for RV volumes and RVEF, compared with CMR reference, and have also demonstrated its additional diagnostic and prognostic value over conventional 2DE parameters [13,14,15,16,17].

However, the main ideal features for all ultrasound measurements are simplicity, quickness and integration into the standard comprehensive transthoracic examination and, in this regard, 2D and 3D evaluation of the RV has had many technical limitations and recent evolutions. As such, 3D moved from complex tracings of the RV in three main derived axes in diastole and systole (coronal, sagittal and frontal cross-sections) to semi-automated methods and from offline to on-board software [10]. 

In this study, we compared the traditional offline 3D software for the analysis of the RV to an on-board AI-based software that Genovese et al. [11] first tested. Previous studies with the use of AI-based approaches (e.g., including machine learning algorithms) showed that this AI methodology enables detection of LV and left atrial boundaries throughout the cardiac cycle from 3D datasets, allowing rapid and accurate measurements of left-side volumes and function [8,9,10,11,12,13,14,15,16,17,18,19,20,21,22]. The same AI methodology has been further developed for endocardial borders of the RV, providing 3D RV volumes and function. This new software potentially overcomes the limitations of previous 3D measurements. Indeed, despite Tamborini et al. [23] and Maffessanti et al. [24] showing a high feasibility and providing reference values for RV 3D volumes with a commercially available dedicated system, a new, more agile method was desired. Moreover, a comprehensive evaluation of RV function includes also M-mode (TAPSE), 2D (FAC) and strain data and this new automated AI-based software allowed 3D calculations and multiparametric 2D and Diffusion Tensor Imaging (DTI) analysis. Interestingly, this automated AI-based system was highly feasible during irregular cardiac rhythms, such as atrial fibrillation (100%), as already demonstrated for LV analysis [25]. 

Therefore, the main novelty of our study is the demonstration that this new automated AI-based system allows a comprehensive assessment of 3D volumes, RVEF and other M-Mode, 2D and strain functional parameters easily, rapidly and accurately, in a variety of pathologies. In this regard, correlations between the 3D traditional semi-automated method (studied in detail in the last few decades in terms of diagnosis, prognosis and comparison with CMR) facilitates our conclusions. As a matter of fact, our data, in a large, unselected population, not only propose this new method for a rapid comprehensive assessment of the RV, but also confirm the utility of a system that produces measurements similar to the established data obtained with the semi-automatic method (including the reference values). In this regard, the comparison between the automated AI-based method with the previous semi-automatic 3D method was pivotal (largely evaluated and correlated to CMR). Moreover, differently from other studies, we also clearly demonstrated that TAPSE, FAC and FWLS could be derived by the same single acquisition of the 3D dataset.

These observations further complete recent data from Namisaki et al. [26]. These authors compared, in a large population, an RV 3D dataset from the focused RV view or the four-chamber view using an automated software, showing similar RV EF and demonstrating significant association with outcomes. Their feasibility was very high (92%) and similar to our data (94.5%) and RVEF from both views with automated analysis did not differ from RVEF measured using CMR. In our study, as we previously demonstrated with the semi-automated system, the transducer was positioned to obtain a 2D RV focused view (generally with a more lateral position of the transducer). Independently of the precise position of the probe, 3D datasets should contain the entire RV and, specifically, the RV free wall should be well defined. Otani et al. [27] compared this new automated software to CMR in 100 cases (and also to the semi-automated 3D system). Automated 3DE RV quantification software underestimated RV volumes but successfully approximated RVEF when compared with CMR. No inferiority in this software was observed when compared with the semi-automated software. Similar to our data, the feasibility and correlation between the automatic and semi-automatic system were generally high. As concerns underestimation of 3D echo volume versus CMR and excellent correlation of RVEF with the two modalities, these are well known data concerning either the LV or the RV volumes. However, Muraru et al. [5] reported that the latest version of semi-automated vendor-independent 3DE RV quantification software and the automated system had an excellent accuracy (bias for RVEDV, −15 mL; bias for RVEF, 1.4%) and reproducibility compared with CMR. Genovese et al. [11] also demonstrated that results obtained from the 3D automated AI-based analysis, with or without the manual correction, had an excellent accuracy (bias for RVEDV, −26 mL; bias for RVEF, 3.3%) when compared with CMR as a reference. Interestingly, similar to this paper, we also showed that editing of the RV contours was necessary in approximately 2/3 of the cases without a significant increase in the time analysis. 

The present study has some limitations. First of all, the absence of a comparison between 3DE and CMR measurements; however, previous studies demonstrated a good correlation between 3DE and CMR RV volumes in a selected population [5,11,15,17,26,27]. The aim of this study was to evaluate the feasibility of the recent 3D automated AI-based method in a large population and our 3DE results were similar to those reported in CMR studies. Moreover, independently of the correlation with CMR studied in depth in the last decade, this new 3DE method clearly facilitates the multiparametric evaluation of the RV, and the potential underestimation of RV volumes with optimal correlation for the RVEF does not detract much from our conclusions. A second limitation is the limited number of patients with abnormal RV volume or complicated congenital heart disease (20 patients). A prospective study with a selected population of patients with congenital pathologies associated with RV pressure or volume overload would reinforce our results. 

## 5. Conclusions

Echocardiographic 3D evaluation of the RV may be included in the standard protocol of transthoracic echocardiography or utilized only in selected cases. Several studies showed that 3D RV assessment is particularly useful in heart failure, congenital heart diseases, pulmonary hypertension, post-cardiac surgery and prior to LV assist device implantation. Obviously, it is also potentially useful in all pathologies, but probably in a busy laboratory, it may be included in the standard examination protocol on the basis of the clinical question.

These new data showed that rapid analysis and excellent reproducibility of 3D automated AI-based RV analysis supported the routine adoption of this method in the daily clinical workflow.

## Figures and Tables

**Figure 1 jcm-11-04528-f001:**
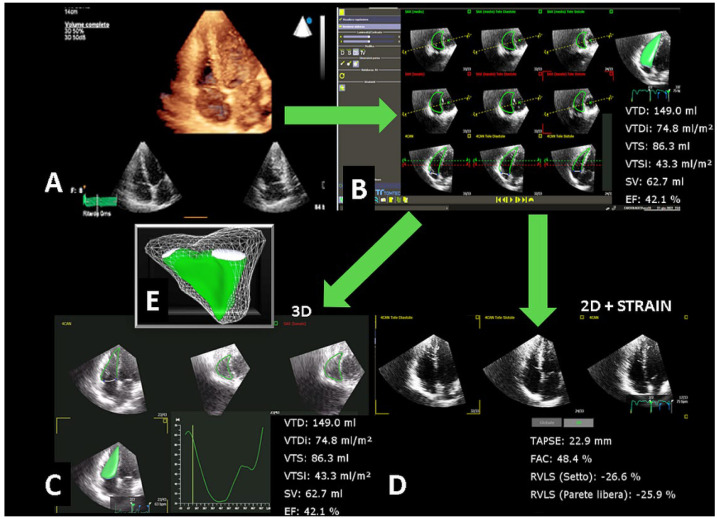
Step–by–step analysis of the RV: (**A**) Acquisition of the full volume from the 4–chamber apical view focused on the RV; (**B**) after acquisition the system is launched and in apparently 14 s the complete 3D analysis is obtained; in the same panel 3D–2D views allow editing of the borders in case of suboptimal contour; (**C**,**D**) by clicking the bottom a 3D analysis with functional curves or a 2D and strain measurements may be easily displayed; (**E**) the 3D dynamic model of the inflow, outflow and apex of the RV may be rotated and displayed freely.

**Figure 2 jcm-11-04528-f002:**
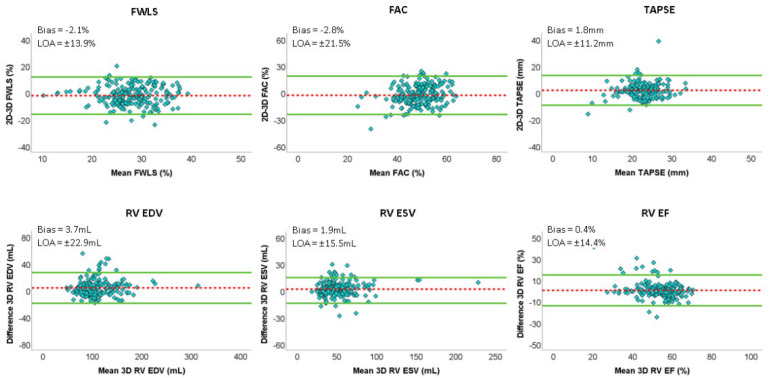
Top panels: results of Bland–Altman analysis for right ventricular free wall longitudinal strain (FWLS), right ventricular fractional area change (RV FAC) tricuspid annular plane systolic excursion (TAPSE) obtained from automatic and semi–automatic 2–dimensional methods. Bottom panels: Results of Bland–Altman analysis for right ventricular end–diastolic volume (RV EDV), right ventricular end–systolic volume (RV ESV) and right ventricular ejection fraction (RV EF) obtained from automatic and semi-automatic methods. Dashed line = bias; solid line = ±2 standard deviations; LOA: limits of agreement.

**Figure 3 jcm-11-04528-f003:**
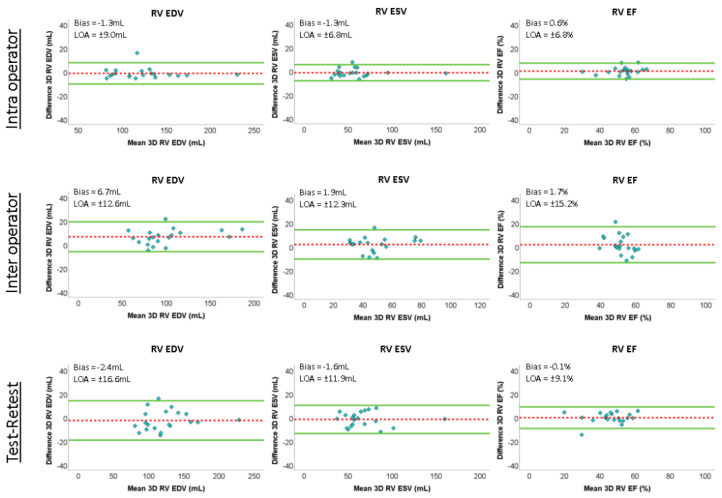
Results of Bland–Altman analysis of the agreement between repeated measurements of right ventricular end–diastolic volume (RV EDV), right ventricular end–systolic volume (RV ESV) and right ventricular ejection fraction (RV EF) for intra–operator (top), inter–operator (middle) and test–retest (bottom) comparisons. Dashed line = bias; soldi line = ±2 standard deviations; LOA: limits of agreement.

**Figure 4 jcm-11-04528-f004:**
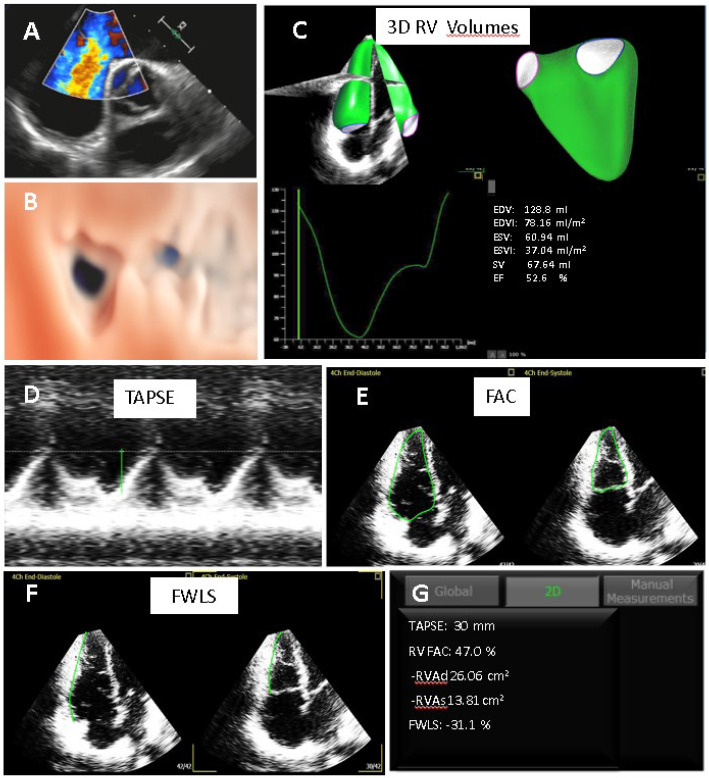
Patient with an ostium secundum ASD. (**A**) TEE showing a significant left–to–right shunt at the atrial level. (**B**) oval shape ASD at transthoracic 3D echo (right atrial view); (**C**) AI 3DTTE volume and function curves and analysis demonstrating moderate dilation of the RV normal RV systolic function. (**D**–**G**) automated analysis of TAPSE, FAC and RVLS obtained by 3D dataset. ASD: atrial septal defect; TEE: transesophageal echocardiography; TTE: transthoracic echocardiography; TAPSE: tricuspid annular plane systolic excursion; FAC: fractional area change; FWLS: free wall longitudinal strain.

**Figure 5 jcm-11-04528-f005:**
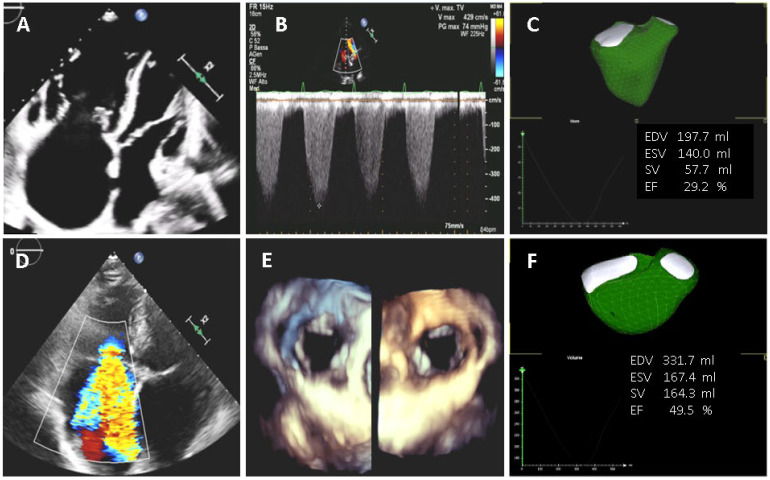
Patient with severe pulmonary hypertension (top panels) and a case with severe volume overload due to tricuspid regurgitation (bottom panels). (**A**) four–chamber view showing severe RV dilation; (**B**) continuous wave Doppler of tricuspid regurgitation indicating severe pulmonary hypertension; (**C**) AI 3D analysis confirms severe dilatation and dysfunction of the RV; (**D**) severe tricuspid regurgitation; (**E**) 3D RV view of the tricuspid valve; (**F**) AI 3D measurements showing RV dilation with normal RV systolic function.

**Table 1 jcm-11-04528-t001:** Demographic and two-dimensional echocardiographic characteristics of study population.

	Total	Normal Subjects	Pathological Patients
Patients (F/M)	203 (86/117)	122 (54/68)	81 (31/50)
Age (years)	57 ± 15	52 ± 14	65 ± 15 **
Body surface area (m^2^)	1.86 ± 0.22	1.87 ± 0.22	1.85 ± 0.21
Left ventricular end diastolic volume (mL)	105.3 ± 40.0	95.1 ± 24.5	120.8 ± 52.3 **
Left ventricular end-systolic volume (mL)	44.5 ± 33.2	36.7 ± 11.5	56.2 ± 48.4 **
Left ventricular end diastolic volume index (mL/m^2^)	56.3 ± 19.8	50.7 ± 10.8	64.8 ± 26.3 **
Left ventricular end-systolic volume index (mL/m^2^)	23.7 ± 17.0	19.5 ± 5.5	30.0 ± 24.9 **
Left ventricular stroke volume (mL)	61 ± 17	58 ± 16	64 ± 16 *
Left ventricular stroke volume index (mL/m^2^)	33 ± 9	31 ± 8	35 ± 9 *
Left ventricular ejection fraction (%)	60 ± 10	62 ± 7	58 ± 14 *
Left atrial diameter (mm)	36.5 ± 5.9	35.7 ± 5.7	38.1 ± 6.0 *
Left ventricular volume (mL)	56.1 ± 25.5	48.7 ± 16.8	67.2 ± 31.8 **
Left ventricular volume index (mL/m^2^)	30.3 ± 15.0	26.1 ± 8.8	36.7 ± 19.5 **
Right ventricular end diastolic area (cm^2^)	17.7 ± 4.6	17.2 ± 4.0	18.3 ± 5.3
Right ventricular end-systolic area (cm^2^)	9.2 ± 3.2	8.9 ± 2.5	9.8 ± 4.1
Right ventricular fractional area change (%)	47.6 ± 9.7	47.8 ± 8.6	47.2 ± 11.2
Tricuspid annular plane systolic excursion (mm)	23.7 ± 5.0	24.3 ± 4.2	22.8 ± 6.1
Right atrial area (cm^2^)	15.8 ± 7.4	14.2 ± 3.6	17.7 ± 10.1 *
Systolic pulmonary artery pressure (mmHg)	30.6 ± 6.7	29.4 ± 3.8	32.4 ± 9.4 *

* = Significant difference *p* < 0.05 vs. normal subjects. ** = Significant difference *p* < 0.001 vs. normal subjects.

**Table 2 jcm-11-04528-t002:** Comparison between 3DE RV parameters evaluated with automatic and semi-automatic methods.

	Automatic	Semi-Automatic	Bias	LOA	ICC
**Total**					
Right ventricular end diastolic volume (mL)	111.4 ± 34.4	107.6 ± 33.4 **	3.7	22.9	0.966
Right ventricular end-systolic volume (mL)	51.7 ± 23.6	50.0 ± 22.5 **	1.9	15.5	0.968
Right ventricular ejection fraction (%)	54 ± 8	54 ± 10	0.4	14.4	0.805
**Normal subjects**					
Right ventricular end diastolic volume (mL)	105.7 ± 28.5	104.3 ± 28.4	1.4	20.4	0.965
Right ventricular end-systolic volume (mL)	47.1 ± 15.9	45.8 ± 15.6	1.2	15.5	0.932
Right ventricular ejection fraction (%)	56 ± 7	56 ± 8	−0.4	13.4	0.723
**Pathological patients**					
Right ventricular end diastolic volume (mL)	120.1 ± 40.6	112.8 ± 39.6 **	7.3	24.9	0.966
Right ventricular end-systolic volume (mL)	58.8 ± 30.7	56.0 ± 29.1 *	2.8	15.6	0.98
Right ventricular ejection fraction (%)	52 ± 9	50 ± 12	1.6	15.6	0.832

LOA: limits of agreement; ICC: intraclass correlation coefficient. * = Significant difference *p* < 0.05 vs. automatic. ** = Significant difference *p* < 0.001 vs. automatic.

**Table 3 jcm-11-04528-t003:** Correlations between 3DE automatic derived and traditional 2DE right ventricular functional parameters.

	Automatic	2D	Bias	LOA	ICC
**Total**					
Right ventricular free wall longitudinal strain (%)	29 ± 6	27 ± 6 **	−2.1	13.9	0.479
Right ventricular fractional area change (%)	50.3 ± 7.8	47.6 ± 9.7 **	−2.8	21.5	0.359
Tricuspid annular plane systolic excursion (mm)	21.9 ± 4.4	23.7 ± 5.1 **	1.8	11.2	0.424
**Normal subjects**					
Right ventricular free wall longitudinal strain (%)	29 ± 6	27 ± 6 **	−2.2	14.2	0.288
Right ventricular fractional area change (%)	51.6 ± 6.8	47.8 ± 8.6 **	−3.8	20.8	0.112
Tricuspid annular plane systolic excursion (mm)	22.2 ± 4.0	24.3 ± 4.2 **	2.1	11.2	0.062
**Pathological patients**					
Right ventricular free wall longitudinal strain (%)	28 ± 7	26 ± 7 *	−2	13.7	0.633
Right ventricular fractional area change (%)	48.4 ± 8.8	47.2 ± 11.3	−1.2	22.2	0.546
Tricuspid annular plane systolic excursion (mm)	21.3 ± 5.0	22.8 ± 6.1 *	1.4	11.1	0.637

LOA: limits of agreement; ICC: intraclass correlation coefficient. * = Significant difference *p* < 0.05 vs. automatic. ** = Significant difference *p* < 0.001 vs. automatic.

## Data Availability

Not applicable.

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
