# Peer review of "Comparison between Automatic and Semiautomatic System for the 3D Echocardiographic Multiparametric Evaluation of RV Function and Dimension"

_jcm, 2022, doi:10.3390/jcm11154528_

Round 1
Reviewer 1 Report
The submitted manuscript titled “Head-to-Head Comparison between Automatic and Semiauto-2 matic System for the 3D Echocardiographic Multiparametric Evaluation of RV Function and Dimension”compared the feasibility and accuracy of right ventricular function analysis of 3D echocardiography utilizing new AI capabilities with conventional 3D and 2D echocardiography. It concluded that the new 3D software is comparable to conventional methods, reduces examination time, and can be used in routine clinical practice. While the focus of this study is interesting and the manuscript is well written, I would like to highlight the following key concerns that need to be addressed to improve the quality of this article.
1. There are no gold standard methods to assess right ventricular volume and function, including MRI.
2. This study is weak in novelty.
3. Image quality should be considered with respect to the accuracy of 3D echocardiography.
4. The authors need to provide more details as to how AF was deal with.
5. What is the level of agreement on the diagnosis of impaired RV function or RV enlargement?
6. Is this new soft ware useful for complicated congenital heart disease?
7. Is the left ventricle not measured in 3D? Also, is the stroke volume measured by Doppler or 2D?
8. A detailed description of how the inspection time was measured needs to be provided in the Methods section.
Reviewer 2 Report
AI- based automated 3DE software for RV evaluation was compared with semiautomatic 3DE method . AI method was faster than 2DE and semi-automatic 3D. But title is missleading "head to head comparison between " - may suggest head to head comparison between authors ? or patients ", so I would prefer to delete unnecessary words and live just "Comparions between". Second point: please provide additional figures (at least 2) with abnormal cases, to show your method and explanation
Round 2
Reviewer 1 Report
The submitted paper was well revised. There are no further comments. Thank you for your revision.